# Genetic Diversity of *Salmonella* Derby from the Poultry Sector in Europe

**DOI:** 10.3390/pathogens8020046

**Published:** 2019-04-04

**Authors:** Yann Sévellec, Arnaud Felten, Nicolas Radomski, Sophie A. Granier, Simon Le Hello, Liljana Petrovska, Michel-Yves Mistou, Sabrina Cadel-Six

**Affiliations:** 1Laboratory for Food Safety, Université PARIS-EST, ANSES, F-94701 Maisons-Alfort, France; yann.sevellec@anses.fr (Y.S.); arnaud.felten@anses.fr (A.F.); nicolas.radomski@anses.fr (N.R.); sophie.granier@anses.fr (S.A.G.); michel-yves.mistou@anses.fr (M.-Y.M.); 2Institut Pasteur, Centre National de Référence des *Salmonella*, Unité des Bactéries Pathogènes Entériques, 75015 Paris, France; lehello-s@chu-caen.fr; 3Animal and Plant Health Agency, Addlestone KT15 3NB, Surrey, UK; Liljana.Petrovska@apha.gsi.gov.uk

**Keywords:** *Salmonella* Derby, reference genome ST71, poultry sector, human infection, phylogenetic analysis, antimicrobial resistance, phage analysis

## Abstract

*Salmonella* Derby (*S*. Derby) is emerging in Europe as a predominant serovar in fattening turkey flocks. This serovar was recorded as being predominant in the turkey sector in 2014 in the United Kingdom (UK). Only two years later, in 2016, it was also recorded in the turkey and broiler sectors in Ireland and Spain. These *S*. Derby isolates were characterised as members of the multilocus sequence type (MLST) profile 71 (ST71). For the first time, we characterise by whole genome sequencing (WGS) analysis a panel of 90 *S.* Derby ST71 genomes to understand the routes of transmission of this emerging pathogen within the poultry/turkey food trade. Selected panel included strains isolated as early as 2010 in five leading European g countries for turkey meat production. Twenty-one of the 90 genomes were extracted from a public database—Enterobase. Five of these originated from the United States (*n* = 3), China (*n* = 1) and Taiwan (*n* = 1) isolated between 1986 and 2016. A phylogenomic analysis at the core-genome level revealed the presence of three groups. The largest group contained 97.5% of the European strains and included both, turkey and human isolates that were genetically related by an average of 35 ± 15 single nucleotide polymorphism substitutions (SNPs). To illustrate the diversity, the presence of antimicrobial resistance genes and phages were characteised in 30, *S*. Derby ST71 genomes, including 11 belonging to this study This study revealed an emergent turkey-related *S*. Derby ST71 clone circulating in at least five European countries (the UK, Germany, Poland, Italy, and France) since 2010 that causes human gastroenteritis. A matter of concern is the identification of a *gyrA* mutation involved in resistance to quinolone, present in the Italian genomes. Interestingly, the diversity of phages seems to be related to the geographic origins. These results constitute a baseline for following the spread of this emerging pathogen and identifying appropriate monitoring and prevention measures.

## 1. Introduction

The European Union (EU) is a major producer of turkey meat worldwide with 80% of production supplied by six countries: Germany, France, Poland, Italy, Spain, and the United Kingdom (UK). Since 2014 the UK has consistently reported high occurrence of *Salmonella enterica* subsp. *enterica* serovar Derby (*S*. Derby) in the turkey sector compared to other countries [1,2,3]. In the EU, in 2016, *S*. Derby was identified in 64.4% of the pig isolates and 32% of the poultry isolates distributed in 21.0% from turkeys and 11.3% from broilers. The most frequent serovar detected in fattening turkey flocks was *S*. Derby (21.8%). Multidrug resistance was reported in 35.7% of the 143 *S*. Derby isolates collected by three European countries (the UK, Ireland, and Spain). Extremely high levels of resistance (>70%) to sulfamethoxazole and tetracycline were detected. Resistance to fluoroquinolone was detected in the UK and Spain [4]. According to the literature [5,6], *Salmonella* Derby isolates from the poultry sector belong mainly to the multilocus sequence type (MLST) profile 71 (ST71). The link between the ST71 and the avian sector can also be appreciated by consulting the freely accessible Enterobase database [7]. Of the 1630 recorded *S.* Derby genomes in Enterobase (on the 1 June 2018), the *S.* Derby ST71 (*n* = 155) were mostly from avian (42%) and human (26%) origin (Appendix A). It must be noted that the *S.* Derby ST71 genomes in Enterobase were mostly recorded between 2014 and 2018 (87%) with the majority (85%) originating from Europe.

Among the *S*. Derby from Enterobase, 1092 genomes belong to the multilocus sequence type (MLST) profile 40 (ST40) (67%), the main MLST profile isolated from the pork sector [8,9]; 7% of these genomes were from avian origins (*n* = 80). These data underline the host specificity of ST40 and ST71 to pork and poultry, respectively. The other 23.5% recorded *S*. Derby from Enterobase are divided between 4 MLST profiles, the ST39 (5%), ST72 (4%), ST682 (13.5%), and ST3871 (1%). The ST39 profile isolated mostly in Germany and France [8,10], is predominantly associated with pork (*n* = 40/78). The ST682 profile has been isolated in Europe and North America. ST682 is associated with a national outbreak in Germany that occurred in 2013 [11]. For the two other MLST profiles, the metadata recorded in Enterobase showed that the ST72 was only recorded in North America and the ST3871 only in Europe.

Considering some human infections have been linked to the consumption of pork meat contaminated with *S.* Derby [8,9,10,11], the emergence of a clone in the turkey and broiler sectors is a matter of concern. Indeed, *S*. Derby ranks among the top eight most commonly reported serovars in humans in the EU and poultry meat is considered as the principal *Salmonella* source of human infection [1,2,3]. According to data recorded by the World Health Organization between 2007 and 2015, the burden of pathogens commonly transmitted through the food chain has increased worldwide and this global rise has been associated mainly with the increase in consumption of products of animal origin [12]. Among these, white meat (pork and broiler meat) play a major role due to increased consumption in line with population growth and the economic crisis [13].

This study aimed to genetically characterise the human and poultry *S*. Derby ST71 strains circulating in Europe and better understand the spread of this emerging pathogen. We compared by variant calling and phylogenomic analyses 90 *S.* Derby ST71 genomes mostly isolated from 2013 to 2016 (92%) in Europe (94%). We compared genomes from turkey (*Meleagris gallopavo*) isolated in the UK by the Animal and Plant Health Agency (AHPA) with ones from guinea fowl, duck, *Gallus gallus*, and turkey isolated in France by the Food Safety Agency (ANSES). We also included the genomes from *S*. Derby ST71-infected-patients isolated by the Institut Pasteur—National Reference Center for *Salmonella* (NRC) in France, and genomes from both, the poultry sector and humans collected from Enterobase that were isolated in other European countries (Germany, Italy, and Poland). Based on the epidemiological data and the pairwise SNP differences we selected 30 genomes representative of the observed diversity of the *S*. Derby MLST profile 71. In order to identify potential genomic differences within the *S.* Derby genomes, we performed phylogenomic reconstruction and investigated the profiles of phages and the acquired antimicrobial resistance genes or mutations conferring antimicrobial resistance.

## 2. Results

The 90 analysed *S*. Derby ST71 genomes were closely related, with an average of 82 single nucleotide polymorphism substitutions (SNPs) and a standard deviation (SD) of 64 SNPs even though the isolates came from different continents and origins (i.e., humans, animals, and food). Nevertheless, the phylogenetic reconstruction showed that different groups could be identified within this lineage as shown in Appendix A. Because of the high clonality of the 59 genomes from the ANSES collection [8]—13 SNPs ± 6 SNPs—, only six genomes were selected for further genomic analysis. Moreover, within the 21 genomes from Enterobase, 7 were excluded due to a lack of metadata. The description of the subset of 30 genomes selected to represent the *S*. Derby ST71 diversity at European and international levels is presented in Table 1.

The phylogenetic analysis of the 30 genomes, genetically distant by 69 SNPs ± 76 SNPs, identified three different groups within this lineage, with the largest group containing 93% of the European strains (*n* = 25/27) (Figure 1). The metrics related to genome assembly, the matrix of pairwise SNP distances and significant differences between the considered distributions of pairwise SNP differences are presented in Appendix A. Kolmogorov-Smirnov tests confirmed statistically significant difference (*p* = 6.66 × 10^−5^) in the distributions of pairwise SNP differences between the group containing majority of the European samples compared to the other two groups. The UK *S*. Derby genomes were genetically related to the strains isolated in the other European countries with an average of 35 SNPs and a standard deviation (SD) of 15 SNPs (Figure 1).

The complete and partial antimicrobial resistance genes identified in the 30 studied *S*. Derby genomes are listed in Appendix A. No resistance genes weredetected in 60% (18/30) of the 30 studied genomes. Thirty-three percent (10/30) could be classified as being multidrug resistant, harboring more than three different resistance mechanisms acquired either by point mutations or gene acquisitions. The genes mediating resistance to aminoglycosides, sulfonamides, and tetracyclines were the most frequently encountered. A S83Y *gyrA* substitution, previously described to mediate resistance to fluoroquinolones [14], was detected in four poultry-related genomes from strains isolated in Italy. The two genomes from the USA did not display known antimicrobial resistance mechanisms. The isolate from Taiwan (BCW 2672) and one French human isolate (291407219) displayed the same genes involved in resistance to aminoglycosides, beta-lactams, sulfonamides, tetracyclines, and phenicols (Figure 2).

The results obtained with the PHASTER software, a phage search tool, showed a total of 10 intact prophages and 10 different integrase genes. It must be noted that PHASTER was unable to identify any integrase genes in some prophages and identified several in others (Appendix A and Figure 2).

## 3. Discussion

Although more strains from the United States and Asia would reinforce our conclusion, the study suggests that the three identified ST71 Derby groups were in accordance with the geographic distribution of the strains (Figure 1). The first group included two strains from the USA, the second grouped one strain from Taiwan and two from French patients, one of whom reported travel to Thailand, and the third included only strains originating from Europe (France, Germany, Poland, Italy, and the UK). Herein, the ST71 *S*. Derby isolated from turkey in the UK by the AHPA in 2016 and notified in the European Food Safety Authority (EFSA) report [3] were genetically related to the strains isolated in the other EU member states (France, Germany, Poland, and Italy). The analyzed European genomes were related with an average of 35 SNPs and a standard deviation (SD) of 15 SNPs. The most genetically distant strain was H122800357 isolated in the UK in 2012 from human (48 SNPs with an SD of 12 SNPs). Our results show that this *S*. Derby ST71 turkey-related clone was circulating in Europe before 2014 (i.e., the year at which it was notified for the first time in the EFSA report) [1] not only in the UK, but also in Poland and Italy (isolates from 2010 and 2013). The two earliest strains in the dataset were isolated in 1986 in the United States from wild birds. These genomes differred by 104 SNPs and an SD of 53 SNPs. The other older genomes in the dataset were one human strain fromTaiwan isolated in 2000 and a poultry strain from Poland isolated in 2010. It is possible that this clone had spread throughout Europe between 2011 and 2017 (Italy, the UK, Germany, and France), via the wild birds, poultry or human transmission routes. The epidemiological data collected seems to support the hypothesis of a transmission of this *S.* Derby ST71 clone from wild bird species to poultry and humans from North America to Asia and Europe during migration, feeding, and wintering. Wild birds and humans have the ability to move over large distances in a short period of time. Many bird species undertake long-distance seasonal movements along migration routes or flyways for breeding, food, and climate purposes [15]., During these movements they have the potential to disperse microorganisms that are potentially harmful to public health and animals. *Salmonella* were found in a wide range of wild birds and furthermore, *S*. Derby has been isolated from gulls (*Larus Ridibundus*), a coastal or inland species feeding and wintering in temperate regions of the globe, such as Europe, East Asia, and North America [16]. The gull-*Salmonella* pathway is considered important in maintaining *Salmonella* in the environment, sustaining transmission to humans and livestock [16,17]. Furthermore, live animals and international food trade could have also contributed to the spread of this clone between countries and especially across Europe.

Owing to the fact that around 36% of the world’s turkey meat production originates from Europe, with only five countries—Germany, France, Poland, the UK, Italy, and Spain—producing 80% of the volume [18], it would have been interesting to also study *S.* Derby strains isolated in Spain. Unfortunately, no Spanish *S.* Derby genomes were available in the free access databases. The UK has been the second leading state in the consumption of white meat after Spain since 2000 [18,19]. Interestingly, the tree genomes, 44675, 1173552, and 198252, isolated in humans in the UK between 2012 and 2015 are genetically distant by only 9 SNPs ± 1.5 SNPs to the genome 17-43350 of a strain isolated from turkey meat in Germany in 2015. In France, *S*. Derby ST71 strains responsible for human infections recorded between 2014 and 2015 by the French National Reference Centre for *Salmonella* represented 2% of the total infections due to *S*. Derby. Among the six patients, two were likely to have contracted the infection while travelling abroad (strains, 201402459 and 201407219, in Figure 1 and Figure 2) as these were phylogenetically closer to the isolate from Taiwan. The remaining four human isolates clustered with the French food strains, 2014LSAL05133 (9 SNPs with an SD of 1), isolated from turkey meat and 2015LSAL02005, 2014LSAL03350, 2014LSAL03694, and 2014LSAL01779 (13 SNPs with an SD of 8) from duck, *Gallus gallus*, and guinea fowl meat, respectively. Albeit the incidence of human gastroenteritis in 2014 and 2015, *S.* Derby ST71 is low in France, and our results confirm the ability of this turkey related clone to infect human. Interestingly, 26% of the *S.* Derby ST71 recorded in Enterobase (last accessed 1 June 2018) are of human origin. Practically, this study demonstrated the presence of the *S*. Derby ST71 clone associated with European turkey meat production that could have been adapted to guinea fowl, duck, and *Gallus gallus*, that has been responsible for human salmonellosis.

While the characterization of this clone by phylogenomic analysis at the core-genome level was able to combine 25 genomes into one ST71 European group, differences between those genomes were also highlighted by the analysis of targeted elements from accessory genomes. Screening antimicrobial resistance genes and phages allowed different genomic signatures that seemed related to geographic or animal species origins to be distinguished.

Interestingly, the genome, 17-43350, from the strain isolated in Germany in 2015, genetically close to the four genomes from the UK human strains (9 SNPs ± 1.5 SNPs), presents no amicrobial resistance gene, conversely to the English ones carrying genes mediating resistance to aminoglycosides, sulfonamides, and tetracyclines. No antimicrobial resistance genes weredetected in the French strains responsible for the four human infections or the five food related strains in this cluster. Nevertheless, the French strain, 2014LSAL02325, isolated from turkey meat carried genes involved in resistance to aminoglycosides, sulfonamides, and tetracyclines. These genes were also carried by the Italian and UK strains isolated from patients and turkey. A matter of concern is that the Italian genomes isolated from turkey also revealed the presence of a S83Y *gyrA* substitution usually involved in resistance to fluoroquinolones. Fluoroquinolone susceptibility tests should be performed to confirm the reduced susceptibility to fluoroquinolones of these strains. Most human *Salmonella* infections are self-limiting gastrointestinal illnesses and do not require any antimicrobial treatment. However, in rare cases, infection may spread and lead to severe enteric disease or invasive infection. In such life-threatening cases, rapid and effective antimicrobial treatment is of utmost importance to prevent poor outcomes in patients. Fluoroquinolones are widely recommended to treat severe *Salmonella* infections in adults [4]. The emergence of a European fluoroquinolone-resistant lineage of the *S*. Derby ST71 in turkey flocks is worrisome. Indeed, it should be noted that 24% of the *S.* Derby strains isolated from fattening turkey flocks in Europe in 2016 were resistant to fluoroquinolones. They were identified in Spain and the UK [4].

The French human strain, 201407219, isolated in 2014 and the strain, BCW_2672, isolated from patients in Taiwan in 2000 shared the same genes conferring resistance to ampicillin, chloramphenicol, streptomycin, sulfamethoxazole, and tetracycline. These two genomes showed the presence of a SGI-1 allele, as already described in *S.* Derby by Beutlich et al. [20], carrying those resistance genes. The strain, 201407219, displayed a partial sequence of the *sul1* gene. Remarkably, those two genomic islands were different from the SGI-1C recently identified in two ST40 French strains isolated from pork meat [21].

Additionally, PHASTER identified the presence of a wide variety of prophages in the food-associated isolates compared to the clinical isolates. We were able to identify three profiles within the European group: The first characterized by phage GF-2 (Edward_GF_2_NC_026611) (France, Poland, and UK genomes), the second by phages GF-2 and SEN34 (Salmon_SEN34_NC_028699) (Germany, Italy, and UK genomes), and the third by phages GF-2 and HP1 (Haemop_HP1_NC_001697) (France genomes). Concerning the turkey isolates, we were able to identify only two of the three profiles described above: The profile, GF-2, characterizing the French and English isolates; and the profile, GF-2 + SEN34, characterizing the German and Italian isolates. The sequence of the GF-2 phage was present in 90% (27/30) of the total genomes analyzed and in all of the European genomes. Historically, Yasuike et al. described for the first time the GF-2 phage, a myovirus lytic bacteriophage in 2015 [22]. It was isolated from a cultured Japanese flounder that succumbed to *Edwardsiella tarda*, a Gram-negative *Enterobacteriaceae* that is the most serious pathogen in both marine and freshwater fish farms worldwide. Since viruses are indeed the most abundant biological entity in aquatic ecosystems [23,24,25,26], it is possible that *Salmonella* Derby would be a natural host of this bacteriophage. Further infectivity tests are needed to confirm this hypothesis. Another strategy could be to screen the clustered regularly interspaced short palindromic repeats (CRISPR)-Cas system (CRISPR *locus*, direct repeats, spacers, and Cas protein) to gain insight as to whether there has been a similar bacterial immune response of *S.* Derby to the exposition to this phage.

Given that Europe covers 36% of the world turkey meat production and that *S*. Derby is known for its ability to infect humans, it is essential that the clones emerging in the European turkey and broiler sectors are characterized. The results of this study show that *S*. Derby ST71 can be divided into at least three different groups based on core-genome analysis. Within this European group, some sub-groups were easily identified through antimicrobial resistance genes (Resfinder) or phages (PHASTER) analysis (accessory genome). This study highlighted that SNP analysis at the core-genome level is an undeniably efficient phylogenomic tool for clustering epidemic or emerging clones. Nevertheless, web-based applications, such as Resfinder and PHASTER can also be used to readily rule out potential differences between strains. Indeed, targeted screening of accessory genomic elements such as antimicrobial resistance genes or phages, enabled the identification of genomic signatures potentially useful for monitoring transmission routes for foodborne microbiological hazards along the food chain. Through this study, the complete closed genome of the European *S*. Derby ST71 poultry-related clone was produced, providing a useful contribution to *S*. Derby understanding. In this last perspective, we are currently exploring the presence of genomic substitutions in the core-genome characteristic of the adaptation of the ST71 and ST40 lineages to poultry and pork, respectively. Adhesion-invasion assays are also under progress to compare the virulence potential of these two *S.* Derby lineages.

## 4. Material and Methods

### 4.1. S. Derby Genomes Selection

Ninety genomes were selected at first. Among these genomes, 4 were from the Animal and Plant Health Agency (APHA) collection, and were isolated in the UK from turkey sector in 2016, and belong to MLST profile 71. Fifty-nine ST71 genomes were from the ANSES collection and were isolated from the poultry sector in France between 2014 and 2015 [21]. Six were from the Institut Pasteur National Reference Center for *Salmonella* collection and were isolated from humans in France. They correspond to six patients infected with *S.* Derby ST71 in the same period. Twenty-one ST71 genomes were from the freely accessible Enterobase (http://enterobase.warwick.ac.uk/) and were isolated from the avian sector and humans, in the UK, Germany, Poland, Italy, the United States, and Asia (China and Taiwan). These last 21 genomes were selected among the 155 available from Enterobase (on the date of 1 June 2018) based on their origin (human and poultry) and assembly availability [7]. Finally, only 6 genomes were selected to represent the ANSES collection [8] because of their high clonality and 14 genomes were selected from Enterobase, excluding ones for which no metadata were available. Information about the subset of 30 genomes selected to represent the *S*. Derby ST71 diversity at European and international levels is listed in Table 1.

### 4.2. Whole Genome Sequencing

Strain 2014LSAL01779 isolated in 2014 from *Gallus gallus* carcasses in Brittany (France) presenting the MLST profile ST71 was sequenced both with Illumina Nextseq (i.e., paired-end read sequencing) and PacBio^®^ (i.e., long read sequencing) technologies, in order to be used as a reference genome for an underrepresented serovar in the free available databases. This genome was used as a reference genome for the downstream analyses. Illumina Nextseq sequencing was performed by the Institut du Cerveau et de la Moelle épinière (ICM) (Pitié-Salpêtrière Hospital, Paris) (www.icm-institute.org) using NextEra XT technology and PacBio^®^ sequencing was performed by Genoscreen (Lille). The quality of the Illumina and PacBio^®^ reads was examined using fastQC V0.11.5 [27]. Prinseq V0.20.4 [28] was used to select Illumina long reads of good quality (no undefined bases, phred > 30, length > 60). The PacBio^®^ reads were assembled using SMRT analysis v2.3.0. In the SMRT analysis, HGAP V3.0 [29] was invoked to correct the sub-reads (length > 1000 bases, read-Score 0.8) and Celera V8.3 [30] was used to perform the assembly (sub-reads length > 500 bases, deep coverage > 25X). Samtools V1.5 [31] was used to map the Illumina short paired-end reads against the PacBio assembly to correct potential assembly mistakes and to determine the depth of the final assembly. The final deep coverage obtained was 140X. A unique contig of 4.86 Mb was obtained with a GC-content of 51.12%. The genome containing 4499 CDS and 90 tRNA, was annotated using Prokka [32].

The 4 genomes from the APHA collection and the 6 from the Institut Pasteur collection were sequenced as previously described by Petrovska et al. [33] and Ung et al., [34] respectively.

***Accession number***. The complete genome sequence of *S. enterica* subsp. *enterica* serovar Derby strain 2014LSAL01779 has been submitted to National Center for Biotechnology Information (NCBI) under accession no. CP026609 with sample identification name (ID) SAMN08470240. The IDs of the other 10 genomes issued from APHA and Institut Pasteur are SAMN09080915, SAMN09080917, SAMEA104448822, SAMEA104448826, SAMEA104448833, SAMEA104448829, SAMN09080916, SAMN09080917, SAMN09080919, and SAMN09080920. Details for each genomic assembly are summarized in Appendix A; the accession codes for each genome are listed in Table 1.

### 4.3. Phylogenetic Analysis

The phylogenomic analysis of the initial panel of 90 genomes and then of the subset of 30 *S*. Derby genomes was performed with the CSIPhylogeny tool provided by the Center for Genomic Epidemiology (CGE) (https://cge.cbs.dtu.dk/services/CSIPhylogeny/) [35] using the default parameters. The assembled genomes were aligned against the *S*. Derby 2014LSAL01779 reference genome (described above) and the SNPs were identified using the mpileup part of SAMTools v. 0.1.18 [36]. Several filters are used by the pipeline to ensure that all SNP positions considered are covered by a minimum amount of reads and are identified with significant confidence with respect to the bases identified at each position. Finally, the phylogenetic analysis was carried out on Fastree V2.1 [37] with maximum likelihood criterion and general time reversible (GTR)-gamma model.

Statistical confidence levels for all topologies were evaluated by the bootstrap method (100 replicates). The phylogenetic tree was visualized using iTOL [38].

### 4.4. Statistical Analyses

The non-normality of the data (i.e., pairwise SNP differences) was checked using the Shapiro test [39] with R from the pairwise matrix generated by the CSIPhylogeny tool described above. Equality of variances for pairwise SNP differences was rejected with the Fisher test [40]. Distributions of pairwise SNP differences were compared with a Kolmogorov-Smirnov tests (KS-test) [41].

### 4.5. Identification of Acquired Resistance Genes

The subset of 30 genomes was analyzed using the ResFinder 2.1 application [42] at the Center for Genomic Epidemiology (CGE) server. The threshold for reporting a match between a gene in the ResFinder database and the input *S*. Derby genome was set at a 90% identity over at least 3/5 of the length of the resistance gene. For strains, 201407219 and BCW_2672, SGI-1 coding sequences (NCBI: AF261825.2) were extracted and blasted against the dataset with the BioNumerics BLAST tool. The complete genomic sequence of SGI-1 was investigated using the BioNumerics alignment and sequence visualization tools.

### 4.6. Detection and Characterization of Phages and Prophages

Each of the 30 assembled genomes chosen to represent the *S.* Derby ST71 genomic diversity was analyzed by PHASTER to identify the presence of prophages and their integrase genes [43]. Only prophages identified as “intact” or “questionable” were considered. The identity of all intact prophage sequences detected by PHASTER was confirmed by BLAST [44,45].

## 5. Conclusions

An emerging *Salmonella* Derby clone characterised by the multilocus sequence type 71 is emerging in turkey and broiler sectors in Europe. This clone was most likely **introduced** in Poland in 2010 through wild birds transmission and has spread from 2011 to 2017 in Italy, the UK, Germany and France through poultry to humans. The genomic characterisation of this clone shows that there is only an average of 35 core-genome substitutions—with a standard deviation (SD) of 15 SNPs—amongst the analysed European genomes even when isolates are with different origin, from different countries and are isolated in different years. The accessory genome allows distinguishing sequence signatures that can be helpful to follow the evolution and spread of this pathogen.

## Figures and Tables

**Figure 1 pathogens-08-00046-f001:**
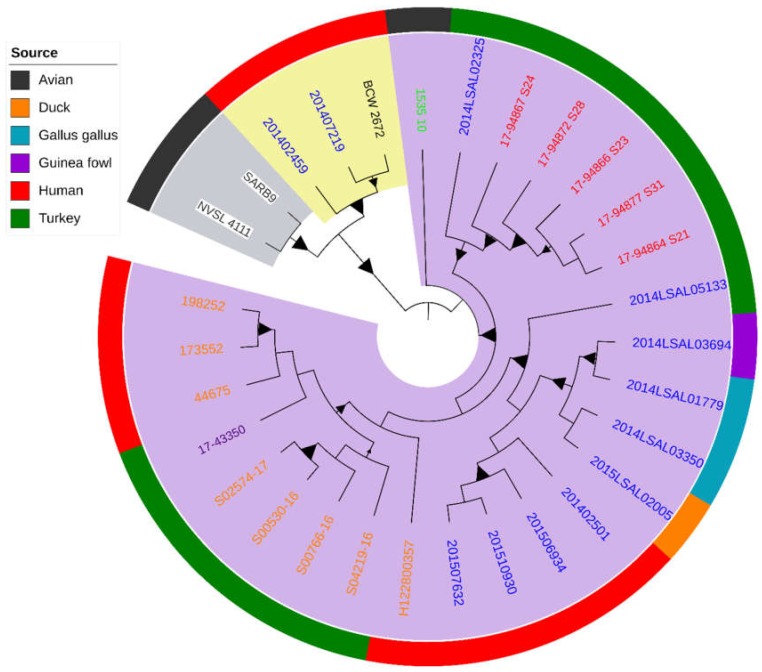
Phylogenetic reconstruction based on core-genome single nucleotide polymorphism substitutions, including strains of the ST71 profile, their geographic distributions and origins. The maximum likelihood criterion and the GTR-gamma model were applied. The branch lengths were not applied. Three groups were identified and the European group is highlighted by a violet colour. The black label indicates strain from Taiwan, blue from France, light green from Poland, orange from the UK, red from Italy, violet from Germany, and white from the United States. Bootstraps comprised between 80% and 100% are shown as triangles at the node’s position.

**Figure 2 pathogens-08-00046-f002:**
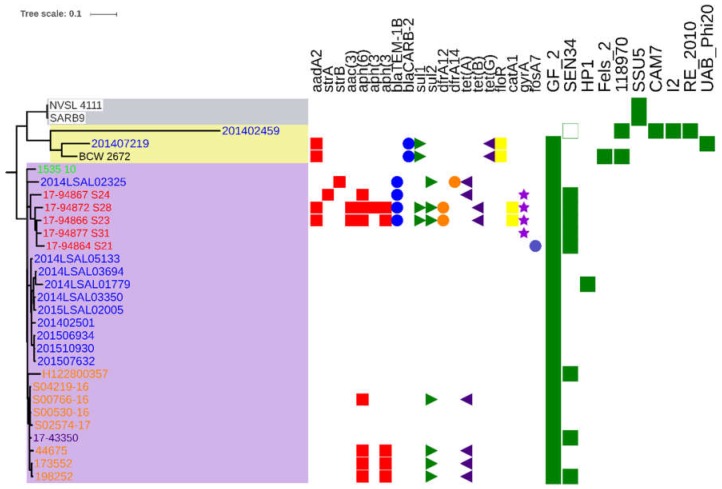
Phylogenetic reconstruction based on core-genome single nucleotide polymorphism substitutions, including strains of the ST71 profile, their geographic distributions, acquired resistance genes, and phage profiles (green boxes). The maximum likelihood criterion and the GTR-gamma model were applied. The scale bar indicates the number of substitutions per site. The black label indicates strains from Taiwan, blue from France, light green from Poland, orange from the UK, red from Italy, violet from Germany and white from the United States. The red cubes correspond to the resistance gene to aminoglycosides antibiotic, the blue circle to beta-lactam, the green triangle to sulfonamides, the orange circle to trimethoprim, the purple triangle to tetracycline, the yellow cube to phenicols, the purple stars correspond to a *gyrA* mutation conferring possible resistance to quinolones and the grey circle to possible resistance to fosfomycines.

**Table 1 pathogens-08-00046-t001:** Information of the 30 *Salmonella* Derby genomes analyzed in this study representing the genomic diversity of the MLST profile 71 in the EU and at e international level. NS = not specified; NA = not available.

Name	Database/Collection	Source Niche	Source Type	Source Details	Collection Year	Continent	Country	Sample ID	Reference
17-43350_S13	Enterobase	Food	Avian	Meleagris; Food, turkey meat	2015	Europe	Germany	SAMEA104379658	NS
17-94867_S24	Enterobase	Poultry	Avian	Meleagris; Animal, animal	2011	Europe	Italy	SAMEA104379733	NS
17-94864_S21	Enterobase	Poultry	Avian	Meleagris; Animal, swab	2011	Europe	Italy	SAMEA104379730	NS
17-94877_S31	Enterobase	Wild animal	Avian	Meleagris; Animal, feces	2013	Europe	Italy	SAMEA104379740	NS
17-94872_S28	Enterobase	Wild animal	Avian	Meleagris; Animal, feces	2012	Europe	Italy	SAMEA104379737	NS
17-94866_S23	Enterobase	Wild animal	Avian	Meleagris; Animal, feces	2011	Europe	Italy	SAMEA104379732	NS
1535_10	Enterobase	Poultry	Avian	NS	2010	Europe	Poland	SAMEA104437530	NS
BCW_2672	Enterobase	Human	Human	Human; *Homo sapiens*	2000	Asia	Taiwan	SAMN02368552	NS
44675	Enterobase	Human	Human	Human; *Homo sapiens*	2014	Europe	UK	SAMN03465613	NS
H122800357	Enterobase	Human	Human	Human; *Homo sapiens*	2012	Europe	UK	SAMN03168996	NS
173552	Enterobase	Human	Human	human; *Homo sapiens*	2015	Europe	UK	SAMN06680388	NS
198252	Enterobase	Human	Human	human; *Homo sapiens*	2015	Europe	UK	SAMN06680393	NS
SARB9 (Fidelma Boyd)	Enterobase	NS	Avian	Bird	1986	North America	United States	NA	NS
NVSL 4111	Enterobase	Wild animal	Avian	NS	1986	North America	United States	SAMN02367710	NS
2014LSAL02325	ANSES	Poultry	Avian	Meat from turkey—carcass	2014	Europe	France	SAMN07734901	[8]
2014LSAL05133	ANSES	Poultry	Avian	Meat from turkey—carcass	2014	Europe	France	SAMN07734953	[8]
2015LSAL02005	ANSES	Poultry	Avian	Meat from duck—fresh	2015	Europe	France	SAMN07734993	[8]
2014LSAL03694	ANSES	Poultry	Avian	Meat from guinea fowl	2014	Europe	France	SAMN07734940	[8]
2014LSAL03350	ANSES	Poultry	Avian	Meat from broilers—*Gallus gallus*—carcass	2014	Europe	France	SAMN07734914	[8]
2014LSAL01779	ANSES	Poultry	Avian	Meat from broilers—*Gallus gallus*—carcass	2014	Europe	France	SAMN08470240	This study
201402459	Institut Pasteur	Human	Human	Human; *Homo sapiens*	2014	Europe	France	SAMN09080915	This study
201407219	Institut Pasteur	Human	Human	Human; *Homo sapiens*	2014	Europe	France	SAMN09080917	This study
S00530-16	APHA	Poultry	Avian	*Meleagris*	2016	Europe	UK	SAMEA104448822	This study
S00766-16	APHA	Poultry	Avian	*Meleagris*	2016	Europe	UK	SAMEA104448826	This study
S02574-17	APHA	Poultry	Avian	*Meleagris*	2017	Europe	UK	SAMEA104448833	This study
S04219-16	APHA	Poultry	Avian	*Meleagris*	2016	Europe	UK	SAMEA104448829	This study
201402501	Institut Pasteur	Human	Human	Human; *Homo sapiens*	2014	Europe	France	SAMN09080916	This study
201506934	Institut Pasteur	Human	Human	Human; *Homo sapiens*	2015	Europe	France	SAMN09080917	This study
201507632	Institut Pasteur	Human	Human	Human; *Homo sapiens*	2015	Europe	France	SAMN09080919	This study
201510930	Institut Pasteur	Human	Human	Human; *Homo sapiens*	2015	Europe	France	SAMN09080920	This study

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
