# Peer review of "Genetic Diversity of Salmonella Derby from the Poultry Sector in Europe"

_pathogens, 2019, doi:10.3390/pathogens8020046_

Round 1
Reviewer 1 Report
Comments to the authors
I read with interest this manuscript. It focuses in the genetic characterization by WGS of Salmonella enterica serovar Derby isolated from poultry and human samples. The results obtained are interesting and of value. Nevertheless, sometimes I found the reading of the manuscript difficult to follow: the connection of ideas, the precision of some aspects, the results section that could be more elaborated and the repetition of ideas. I give my opinion and comments that authors might considered to improve the manuscript.
1. Obejctives. Line 59. Why do the authors aimed to characterize genetically the ST71 S. Derby and compare them with USA and Asia isolates? What is the ultimate and precise objective of this study? To demonstrate the usefulness of WGS as a tool in epidemiology?..... To find genetic markers to profile the isolates epidemiologically ?..... The objectives are not precise.
2. Asia is a big continent. I would say the origin of the Asian isolates (the country) since the authors identified the American and European countries
3. 3. I think that more information should be added in the results section; just an overview of the information given by the tables and figures. For example, it would be nice to read what kind of resistance genes were found in general.
4. Line 75. As the authors state at the discussion, it is premature to delineate a phylogeny with so few isolates from America and Asia.
5. Line 95. Why the authors infer that there are gene acquisitions? Which genes o you refer? Did you study the genetic environment of these genes?
6. Line 97. Why do authors find interesting the two genomes of the USA isolates not showing resistance genes?
7. The discussion is quite large and sometimes is not based in the results obtained. For example, page 8 is almost the description of what is found in the literature and there is no precise connection with the results obtained in this study.
8. Line 117. This first sentence is a conclusion but it misses to say “based on what”. The reader that will only see the discussion will not know what are the markers that lead the authors to this conclusion (if it is correct since the number of isolates of USA and Asia are small…). It is a preliminary (premature) conclusion, I would say (this study suggests ….)
9. Line 166-168. Why this study confirms the ability of these strains to infect humans? Because they came from human samples…. Not because of this study. Actually, I wonder why the virulence genes were not considered in this genetic characterization. It would be very interesting to see if there was some variability in the virulence genes of ST71 from isolates of different origins or geographic locations.
10. Line 176. For example, this sentence should be in the results sections (just to justify my comment above)
11. Line192-195. What is the relevance of including the information of a resistance gene in the middle of a paragraph that elaborates on the presence SGI-1? It is not explained.
12. Lines 218-219. It is true that antimicrobial resistance genes can be more common in one geographic region or more frequently found in isolates from a specific animal due to different politics of the use of antibiotics, which differs from countries and animal-type production. However, many of these resistance genes are part of the accessory genome which may bias this sentence. How can we be sure that these are “genomic signatures”?......
13. Finally, Salmonella enteric is known to infect humans. It is not only S. Derby (line 308). In fact, we do not know the real prevalence of salmonellosis since many infections are not diagnosed at laboratory level, and we do not know exactly the prevalence of the serovars.
14. Line313. I do not agree with this conclusion by the reason of comment 12. The number of the isolates are low and resistance genes often are inserted in genetic mobile elements.
Minor comments:
Line 57. Pork is not white meat. Consider remove “white meat”.
Line 206. Gram (it is the name of a person)
References. They must be edited (journals are in the abbreviated form, sometimes not, etc)
Author Response
Comments and Suggestions for Authors
Comments to the authors
I read with interest this manuscript. It focuses in the genetic characterization by WGS of Salmonella enterica serovar Derby isolated from poultry and human samples. The results obtained are interesting and of value. Nevertheless, sometimes I found the reading of the manuscript difficult to follow: the connection of ideas, the precision of some aspects, the results section that could be more elaborated and the repetition of ideas. I give my opinion and comments that authors might considered to improve the manuscript.
Obejctives. Line 59. Why do the authors aimed to characterize genetically the ST71 S. Derby and compare them with USA and Asia isolates?
Authors: The sentence in line 59 was changed as follow :
“This study aimed to characterise genetically the S. Derby ST71 strains isolated from poultry and humans circulating in Europe in order to better understand the spread of this emerging pathogen.”
In general, in all document, we were attentive to change all formulations concerning genomes from USA and Asia to not let to confusion. We agree indeed that more genomes would be necessary to conclude about comparison of European strain and USA or Asia (as we declare in Discussion, lines 154-159). We also changed the figure in this perspective. All the genomes of S. Derby ST71 available in the free access data bases from USA and Asia were included in this study.
What is the ultimate and precise objective of this study? To demonstrate the usefulness of WGS as a tool in epidemiology?..... To find genetic markers to profile the isolates epidemiologically ?..... The objectives are not precise.
Authors: We clarified the objective of the study in abstract (lines 15-17 and 33-34), introduction (lines 76-77 and 87-88). We reorganized the results and the discussion to better focus on the objective of the study (=circulating strains characterization of an emergent S. Derby clone by WGS analysis)
Asia is a big continent. I would say the origin of the Asian isolates (the country) since the authors identified the American and European countries
Authors: We changed “Asia” by the country (lines 20, 121, 134, 146, 157, 168, 245, 304)
3. 3. I think that more information should be added in the results section; just an overview of the information given by the tables and figures. For example, it would be nice to read what kind of resistance genes were found in general.
Authors: the results were reorganize and more information were added both concerning analysis of genomes and resistance genes (lines 95-100, 105-107, 111-113, 129-131). The table 1 was also moved to better illustrate the work undertaken and the results obtained.
Line 75. As the authors state at the discussion, it is premature to delineate a phylogeny with so few isolates from America and Asia.
Authors: the sentence was changed and all paragraph was reformulated.
5. Line 95. Why the authors infer that there are gene acquisitions? Which genes o you refer? Did you study the genetic environment of these genes?
Authors: we analyzed the presence of genes or mutation by RESFinder. It is concerning the tool used.
6. Line 97. Why do authors find interesting the two genomes of the USA isolates not showing resistance genes?
Authors: the sentence was reformulated (lines 133-134).
7. The discussion is quite large and sometimes is not based in the results obtained. For example, page 8 is almost the description of what is found in the literature and there is no precise connection with the results obtained in this study.
Authors: the discussion was reorganized for a easily reading and following of the idea. Several sentences were reformulated and the long description in page 8 of what was found in the literature reformulated and reduced.
8. Line 117. This first sentence is a conclusion but it misses to say “based on what”. The reader that will only see the discussion will not know what are the markers that lead the authors to this conclusion (if it is correct since the number of isolates of USA and Asia are small…). It is a preliminary (premature) conclusion, I would say (this study suggests ….)
Authors: the sentence was modified following the suggestion of the reviewer. In general all the document was reformulated about USA and Asia isolates. (See also answer to the comment 1)
9. Line 166-168. Why this study confirms the ability of these strains to infect humans? Because they came from human samples…. Not because of this study. Actually, I wonder why the virulence genes were not considered in this genetic characterization. It would be very interesting to see if there was some variability in the virulence genes of ST71 from isolates of different origins or geographic locations.
Authors : This sentence was reformulated for better understanding (lines 207-212 ): “The strains isolated from these four patients clustered with the French food strains 2014LSAL05133 (9 SNPs with an SD of 1), isolated from turkey meat and 2015LSAL02005, 2014LSAL03350, 2014LSAL03694, 2014LSAL01779 (13 SNPs with an SD of 8) from duck, Gallus gallus and guinea fowl meat, respectively. Albeit the incidence of human gastroenteritis in 2014 and 2015 due to this S. Derby ST71 is low in France, our results confirm the ability of this turkey related clone to infect human”.
There is a scientific agreement (literature about Salmonella SNP analysis, Clustering EQA, WGS projects) about the fact that less than 20-15 SNPs = two strains are clonal. In this epidemiological context (temporal and geographical context, for these strains in France), the fact that these strains shear less than 13 SNP substitutions in their core-genome is consistent with link between cause and effect: this food = source of infection for this patient.
In the reformulated sentence, we also developed the idea that this emerging turkey clone is henceforth adapted to different avian reservoirs (Gallus gallus, duck..).
10. Line 176. For example, this sentence should be in the results sections (just to justify my comment above)
Authors: we reformulated the paragraph before this sentence that is now justified in this section of the discussion.
11. Line192-195. What is the relevance of including the information of a resistance gene in the middle of a paragraph that elaborates on the presence SGI-1? It is not explained.
Authors: we reformulated the sentence (lines 247-248): “These two genomes showed the presence of a SGI-1 allele as already described in S. Derby by Beutlich et al. [22] and carrying those resistance genes.”
12. Lines 218-219. It is true that antimicrobial resistance genes can be more common in one geographic region or more frequently found in isolates from a specific animal due to different politics of the use of antibiotics, which differs from countries and animal-type production. However, many of these resistance genes are part of the accessory genome which may bias this sentence. How can we be sure that these are “genomic signatures”?......
Authors: this sentence were delated and this part of the discussion reformulated (lines 270-293)
13. Finally, Salmonella enteric is known to infect humans. It is not only S. Derby (line 308). In fact, we do not know the real prevalence of salmonellosis since many infections are not diagnosed at laboratory level, and we do not know exactly the prevalence of the serovars.
Authors: Conclusions were integrated in the discussion and reformulated (lines 270-293)
14. Line313. I do not agree with this conclusion by the reason of comment 12. The number of the isolates are low and resistance genes often are inserted in genetic mobile elements.
Authors: see the answers in the last two comments.
Minor comments:
Line 57. Pork is not white meat. Consider remove “white meat”.
Authors: maybe it is a cultural definition but in FAO documents white meat include poultry as well as pork. (example is http://www.fao.org/3/y4252e/y4252e07.htm)
Line 206. Gram (it is the name of a person)
Authors: corrected in line 263
References. They must be edited (journals are in the abbreviated form, sometimes not, etc)
Authors: the references were revisited.
Submission Date
05 February 2019
Date of this review
27 Feb 2019 06:59:37

Reviewer 2 Report
Manuscript Number: pathogens-450812
Title: Genetic diversity of Salmonella Derby from poultry 1 sector in Europe
· In the abstract, it was mentioned that 90 genomes were analyzed. Please make clear how many genomes were obtained from databases and other laboratories and how many new genomes were sequenced in the current study.
· In the introduction, please add some info about the various MLST profiles, and identify MLST profile 71; give also examples of other ST71 bacterial species, if any.
· In the results section, the author needs to elaborate more about the SNPs found among the various S. Derby isolates – talk about what genes contain these SNPs, do all of these SNPs contribute to antimicrobial resistance or confer other important biological features to the relevant isolates?, are certain SNPs restricted to certain geographical areas or certain host(s)? etc.
· The phylogenetic analysis needs to involve more American and Asian S. Derby isolates, if available.
· The language needs more attention. Some language errors and typos exist and need corrections. Below are only a few examples.
- Line 13: “…it was also recorded predominantly….” – The meaning is not clear -please rephrase.
- Line 14: “…MLST profile…” Spell out the MLST abbreviation at its first appearance.
- Line 15: Spell out “WGS “ abbreviation
- Line 27: “…..identifying appropriated prevention measures…” – Change “appropriated” to “appropriate”.
- Line 38: “Extremely high levels of resistance…” – replace “Extremely “ with “Very”
- Line 42-44: “With regard to the 1,630 recorded S. Derby from Enterobase the June 1st 2018, the S. Derby ST71 (n=155) are indeed mostly from avian (42%) and human (26%) origin (S1 Table).”
o “Enterobase the June 1st 2018” – not sure what this means!
o Remove “indeed”
- Line 47-49: If the 7% of turkey S. Derby isolates belong to the lineage ST71, then please indicate that.
- Line 67: “….isolated in others European countries…” – change “others” to “other”.
- Line 85: “…coregenome…” – correct misspelling. Also in Line 106.
- Line 136: “…Salmonella were found in a wild range of wild birds….” – Change “wild range” to “wide range”
- Line 313-316: “The complete closed genome of one European S. Derby strain, isolated from poultry, belonging to the genomic lineage ST71 has been produced, providing a significant contribution to the understanding of this serotype for which no complete genome was available until recently.” - The sentence is not very clear and needs rephrasing / rewording.
END
Author Response
Comments and Suggestions for Authors
Manuscript Number: pathogens-450812
Title: Genetic diversity of Salmonella Derby from poultry 1 sector in Europe
· In the abstract, it was mentioned that 90 genomes were analyzed. Please make clear how many genomes were obtained from databases and other laboratories and how many new genomes were sequenced in the current study.
Authors: the abstract was reformulated to better express this point concerning how many new genomes were sequenced in the current study (lines .17-27). Moreover, in the results and M&M the sentences concerning the genomes used were revisited to make it more understandable; the Table1 was also moved up
· In the introduction, please add some info about the various MLST profiles, and identify MLST profile 71; give also examples of other ST71 bacterial species, if any.
Authors: the MLST profile 71 is characteristic of Salmonella Derby. We added a part in the introduction to better explain the diversity of MLST profiles encountered for S. Derby (lines 58-66)
· In the results section, the author needs to elaborate more about the SNPs found among the various S. Derby isolates – talk about what genes contain these SNPs, do all of these SNPs contribute to antimicrobial resistance or confer other important biological features to the relevant isolates?, are certain SNPs restricted to certain geographical areas or certain host(s)? etc.
Authors: we are processing these analyses + cellular test. The results of these analysis will be published soon. To underline the scientific interest of this kind of analysis we reformulated the sentence in lines 289-293
· The phylogenetic analysis needs to involve more American and Asian S. Derby isolates, if available.
Authors: all the USA and Asian isolated available in the free access databases were included in this study. Anyway, we reformulated our conclusions and purposes about USA and Asian isolates taking in account the limited number of isolates analyzed
· The language needs more attention. Some language errors and typos exist and need corrections. Below are only a few examples.
Authors: the English of the article was revisited by a “native”.
- Line 13: “…it was also recorded predominantly….” – The meaning is not clear -please rephrase.
Authors: the sentence was corrected in line 13
- Line 14: “…MLST profile…” Spell out the MLST abbreviation at its first appearance.
- Line 15: Spell out “WGS “ abbreviation
Authors: both these suggestion were integrated (lines 14-15)
- Line 27: “…..identifying appropriated prevention measures…” – Change “appropriated” to “appropriate”.
Authors: done
- Line 38: “Extremely high levels of resistance…” – replace “Extremely “ with “Very”
Authors: done
- Line 42-44: “With regard to the 1,630 recorded S. Derby from Enterobase the June 1st 2018, the S. Derby ST71 (n=155) are indeed mostly from avian (42%) and human (26%) origin (S1 Table).”
o “Enterobase the June 1st 2018” – not sure what this means!
Authors: done
Remove “indeed”
Authors: done
- Line 47-49: If the 7% of turkey S. Derby isolates belong to the lineage ST71, then please indicate that.
Authors: the sentence was reformulated lines 58-66
- Line 67: “….isolated in others European countries…” – change “others” to “other”.
Authors: done
- Line 85: “…coregenome…” – correct misspelling. Also in Line 106.
Authors: done
- Line 136: “…Salmonella were found in a wild range of wild birds….” – Change “wild range” to “wide range”
Authors: done
- Line 313-316: “The complete closed genome of one European S. Derby strain, isolated from poultry, belonging to the genomic lineage ST71 has been produced, providing a significant contribution to the understanding of this serotype for which no complete genome was available until recently.” - The sentence is not very clear and needs rephrasing / rewording.
Authors: this sentence was reformulated lines 288-289
END
Submission Date
05 February 2019
Date of this review
03 Mar 2019 00:18:10
Round 2
Reviewer 1 Report
All the comments were addressed and I find that the manuscript is more precise and clear.
Reviewer 2 Report
Going through the authors' responses to the first review report, it
seems that they have reasonably addressed all the major comments and
concerns. I do not have any further comments. Since the Review Report
Form is not currently available, I cannot answer the questions included
in the form. Please let me know if you have any question.